# Generation of a Transgenic Zebrafish Line for In Vivo Assessment of Hepatic Apoptosis

**DOI:** 10.3390/ph14111117

**Published:** 2021-10-31

**Authors:** Aina Higuchi, Eri Wakai, Tomoko Tada, Junko Koiwa, Yuka Adachi, Takashi Shiromizu, Hidemasa Goto, Toshio Tanaka, Yuhei Nishimura

**Affiliations:** 1Department of Integrative Pharmacology, Mie University Graduate School of Medicine, Tsu 514-8507, Mie, Japan; 320m003@m.mie-u.ac.jp (A.H.); 318d026@m.mie-u.ac.jp (E.W.); j-koiwa@med.mie-u.ac.jp (J.K.); 319m001@m.mie-u.ac.jp (Y.A.); tshiromizu@med.mie-u.ac.jp (T.S.); 2Ise Red Cross Hospital, Ise 516-8512, Mie, Japan; tomoko.tada0426@gmail.com; 3Department of Histology and Cell Biology, Mie University Graduate School of Medicine, Tsu 514-8507, Mie, Japan; hidegoto@med.mie-u.ac.jp; 4Department of Systems Pharmacology, Mie University Graduate School of Medicine, Tsu 514-8507, Mie, Japan; tanaka@doc.medic.mie-u.ac.jp

**Keywords:** liver, apoptosis, caspase, Förster resonance energy transfer, zebrafish, in vivo fluorescence imaging, drug-induced liver injury, hepatoprotectant

## Abstract

Hepatic apoptosis is involved in a variety of pathophysiologic conditions in the liver, including hepatitis, steatosis, and drug-induced liver injury. The development of easy-to-perform and reliable in vivo assays would thus greatly enhance the efforts to understand liver diseases and identify associated genes and potential drugs. In this study, we developed a transgenic zebrafish line that was suitable for the assessment of caspase 3 activity in the liver by using in vivo fluorescence imaging. The larvae of transgenic zebrafish dominantly expressed Casper3GR in the liver under control of the promoter of the phosphoenolpyruvate carboxykinase 1 gene. Casper3GR is composed of two fluorescent proteins, tagGFP and tagRFP, which are connected via a peptide linker that can be cleaved by activated caspase 3. Under tagGFP excitation conditions in zebrafish that were exposed to the well-characterized hepatotoxicant isoniazid, we detected increased and decreased fluorescence associated with tagGFP and tagRFP, respectively. This result suggests that isoniazid activates caspase 3 in the zebrafish liver, which digests the linker between tagGFP and tagRFP, resulting in a reduction in the Förster resonance energy transfer to tagRFP upon tagGFP excitation. We also detected isoniazid-induced inhibition of caspase 3 activity in zebrafish that were treated with the hepatoprotectants ursodeoxycholic acid and obeticholic acid. The transgenic zebrafish that were developed in this study could be a powerful tool for identifying both hepatotoxic and hepatoprotective drugs, as well as for analyzing the effects of the genes of interest to hepatic apoptosis.

## 1. Introduction

Hepatic apoptosis is associated with liver damage that is linked to a variety of etiologies, including viral infection, excess alcohol intake, cholestasis, steatosis, and medication toxicity [1,2,3]. For example, a hepatitis C virus infection can activate death receptors and induce the endoplasmic reticulum stress response in liver cells, resulting in the activation of extrinsic and intrinsic apoptosis pathways, respectively [1,3]. Alcohol (i.e., ethanol) is metabolized by cytochrome P450 2E1 in the liver, which activates the intrinsic apoptosis pathway via the production of reactive oxygen species and the release of cytochrome c from the mitochondria [1,3,4]. Alcohol intake can also alter the permeability of the intestines and elevate circulating levels of lipopolysaccharide (LPS), which in turn leads to increased tumor necrosis factor production by Kupffer cells and activation of the extrinsic apoptosis pathway in hepatocytes [1,3,4]. Both the extrinsic and intrinsic pathways activate caspase 3 [1,2,3]. Therefore, it is important to assess caspase 3 activity in order to elucidate the role of apoptosis in liver damage that is associated with a variety of etiologies.

Although several in vitro liver models using human cells have been developed [5,6,7], animal models are still crucial for elucidating the pathophysiologic mechanisms of liver diseases [8,9]. Increased caspase 3 activity in mice has been successfully monitored using techniques such as bioluminescence imaging in ischemia-reperfusion injury [10] and LPS injection [11]. Zebrafish have recently emerged as a useful in vivo model for studying a variety of disorders, including liver diseases [5,12,13,14]. Human and zebrafish livers have several important differences, including at the cellular and tissue architecture levels and with regard to the enzymes involved in xenobiotic metabolism [15,16]. Nevertheless, a wide variety of assays that are useful for analyzing liver function in zebrafish have been developed and utilized in order to study liver diseases and identify hepatotoxicants and hepatoprotectants [5,12,13,14]. For example, quantitative polymerase chain reaction analyses of liver biomarkers such as ceruloplasmin, transferrin, and fatty acid-binding protein 10a (fabp10a) revealed the hepatotoxicity of acetaminophen, amiodarone, and coumarin in larval zebrafish [17]. Oil Red O staining of larval zebrafish has been used to examine steatosis caused by consumption of a high-fat diet [18] and by dexamethasone (DEX) [19]. Transgenic zebrafish expressing a fluorescent protein under control of the liver-specific gene *fabp10a* promoter were used to detect the hepatotoxicity of acetaminophen and the protective effect of prostaglandin E2 against acetaminophen-induced liver toxicity [20]. The knockout of a key antioxidant-responsive gene, nuclear factor erythroid 2-related factor 2a, increased the susceptibility to acetaminophen-induced hepatotoxicity in zebrafish [21]. A study that employed transgenic zebrafish expressing fluorescent proteins suitable for assessing caspase 3 activity by using Förster resonance energy transfer (FRET) imaging, under the control of a ubiquitously expressed gene promoter, detected heavy metal-associated toxicity [22]. In this study, we developed a transgenic zebrafish that selectively expressed a FRET-based fluorescent protein, Casper3GR, in the liver and examined the utility of this transgenic zebrafish model for the assessment of hepatotoxicity and hepatoprotection of drugs, focusing on caspase 3 activity.

## 2. Results

### 2.1. Generation of a Transgenic Zebrafish Expressing Casper3GR in Hepatocytes

Casper3GR [23] was chosen to detect the activation of caspase 3 in zebrafish. Casper3GR is composed of a tagGFP [24] moiety connected to tagRFP [25] via a linker containing the caspase 3-cleavable motif DEVD [26]. The high fluorescence quantum yield of tagGFP, high molar extinction coefficient of tagRFP, and high Förster radius between the tagGFP and tagRFP makes the DEVD linker connecting these fluorescent proteins suitable for FRET analyses in order to monitor the activation of caspase 3 [23]. Analyses of apoptosis using FRET-based imaging have been reported in zebrafish expressing FRET sensors throughout the body [22,27] or in keratinocytes [28]. In this study, we generated a transgenic zebrafish expressing Casper3GR in order to detect apoptosis in the liver using FRET-based in vivo imaging (Figure 1A). We generated a transposon vector containing the coding region of Casper3GR downstream of the phosphoenolpyruvate carboxy kinase 1 (pck1) promoter [29] in order to dominantly express Casper3GR in larval zebrafish livers (Figure 1B). Zebrafish embryos were microinjected with the transposon vector and transposase mRNA, and the larvae expressing Casper3GR in the hepatocytes were then selected. A transgenic zebrafish line dominantly expressing Casper3GR in the liver at four, five, and six days post-fertilization (dpf) (Figure 1C), which we designated Tg (pck1:Casper3GR), was generated by raising and mating these zebrafish.

### 2.2. FRET-Based Imaging of Tg (pck1:Casper3GR) Detects Isoniazid (INH)-Induced Hepatocyte Apoptosis

Tg (pck1:Casper3GR) was exposed to INH in order to assess the capability of detecting hepatic apoptosis. Previous studies using various models, including zebrafish [30] and immortalized hepatocyte cell lines that were derived from humans [31], demonstrated that exposure to high concentrations of INH can induce the apoptosis of hepatocytes. The exposure of zebrafish to 6 mM INH from 4 to 5 dpf significantly increased the ratio of fluorescence-emitting areas that were detected using narrow and wide filters (Figure 2A,B), suggesting that caspase 3 had been activated and digested the DEVD linker between tagGFP and tagRFP, resulting in a decrease in the emission of fluorescence from tagRFP through FRET and an increase in the emission of fluorescence from tagGFP upon excitation at 460–480 nm. We also analyzed caspase 3 activity using a commercially available luminescent assay [32]. We could not detect any difference in luminescence in the extracts of the whole body of the zebrafish at 5 dpf with or without INH treatment (6 mM for 24 h), possibly due to the noise from caspase 3 activity in tissues other than liver (data not shown). We therefore measured the luminescence in the liver that was removed from the zebrafish at one month post-fertilization (mpf). Because INH treatment at 2 mM for 24 h was lethal to the zebrafish at 1 mpf (data not shown), we decided to treat the zebrafish with INH at 0.5 mM for 24 h. After treatment, we removed the liver and measured the caspase activity. As shown in Figure 2C, the luminescence intensity in the liver of the zebrafish at 1 mpf and treated with INH at 0.5 mM for 24 h was higher than that of the zebrafish without INH treatment, suggesting that caspase 3 is activated in the liver of the zebrafish that was treated with INH.

### 2.3. Detection of Ursodeoxycholic Acid (UDCA)-Associated Hepatoprotection Using FRET-Based Imaging of Tg (pck1:Casper3GR)

UDCA is used clinically as a hepatoprotectant, and it has been shown to suppress hepatocyte apoptosis in cases of cholestatic liver injury, such as primary biliary cholangitis and INH- and rifampin-induced hepatotoxicity [33,34,35]. We therefore examined the effect of UDCA on INH-induced hepatocyte apoptosis. Exposure of Tg (pck1:Casper3GR) to UDCA (100 μM and 200 μM) for 24 h (from 4 to 5 dpf) did not significantly affect the ratio of fluorescence-emitting areas, which were detected using narrow and wide filters, compared to that of the control zebrafish (Appendix A). Co-treatment of Tg (pck1:Casper3GR) with UDCA (200 μM) and INH (6 mM) significantly decreased the fluorescence ratio compared to that of Tg (pck1:Casper3GR) treated with INH (6 mM) only, suggesting that UDCA suppresses INH-induced hepatocyte apoptosis (Figure 3).

### 2.4. Detection of Obeticholic Acid (OCA)-Associated Hepatoprotection Using FRET-Based Imaging of Tg (pck1:Casper3GR)

OCA, a synthetic agonist for the farnesoid X receptor (FXR) [34], ameliorates the severity of acute liver injury that is induced by exposure to carbon tetrachloride and INH in rodents [36,37]. We therefore examined the effect of OCA on the prevention of INH-induced hepatocyte apoptosis. Exposure of Tg (pck1:Casper3GR) to OCA (2.5 μM and 25 μM) for 24 h (from 4 to 5 dpf) did not significantly affect the ratio of fluorescence-emitting areas, which were detected using narrow and wide filters, compared to that of the control zebrafish (Appendix A). Co-treatment of Tg (pck1:Casper3GR) with OCA (25 μM) and INH (6 mM) significantly decreased the fluorescence ratio compared with INH treatment (6 mM) alone, suggesting that OCA suppresses INH-induced hepatocyte apoptosis (Figure 4).

### 2.5. Detection of DEX-Induced Hepatotoxicity Using FRET-Based Imaging of Tg (pck1:Casper3GR)

The use of corticosteroids to treat drug-induced liver injury (DILI) has both pros and cons [38]. Therefore, we examined the effect of DEX on INH-induced hepatocyte apoptosis. Exposure of Tg (pck1:Casper3GR) to DEX (10 μM and 100 μM) for 24 h (from 4 to 5 dpf) did not significantly affect the ratio of fluorescence-emitting areas, which were detected using narrow and wide filters, compared to that of control zebrafish (Appendix A). Co-treatment of Tg (pck1:Casper3GR) with DEX (10 or 100 μM) and INH (6 mM) did not significantly decrease the fluorescence ratio compared with INH treatment (6 mM) alone, suggesting that DEX does not suppress INH-induced hepatocyte apoptosis (Figure 5).

## 3. Discussion

Here, we report the development of a transgenic zebrafish line designated Tg (pck1:Casper3GR), which expresses a FRET-based sensor that is suitable for assessing caspase 3 activity in the liver. Our results demonstrate the usefulness of Tg (pck1:Casper3GR) for drug-discovery studies targeting hepatic apoptosis. The main advantage of Tg (pck1:Casper3GR) is that the activity of caspase 3 in the liver can be measured by in vivo fluorescence imaging. The analysis can be performed without laborious procedures such as isolation of the liver from the whole body and preparation of a solution to measure the caspase activity. It should be noted, however, that Casper3GR is expressed in the kidney at 1 mpf in Tg (pck1:Casper3GR) (data not shown), which is consistent with previous reports that demonstrated the expression of Pck1 in both the liver and kidneys of mice [39,40]. We also used an albino line that lacks melanophores, but not iridophores and xanthophores [41]. The albino line may not be suitable for in vivo fluorescence imaging after the start of adult pigment pattern development around 25 dpf [42]. Therefore, larval Tg (pck1:Casper3GR) are suitable for the FRET-based assessment of hepatocyte apoptosis. For the FRET-based assessment of hepatocyte apoptosis in the adult stage, it may be necessary to use a liver-specific promoter such as fabp10a [43] and the Casper line, which is transparent due to the impairment of melanophores, iridophores, and xanthophores [44].

In this study, we used INH as a hepatotoxicant, as it readily induces hepatocyte apoptosis [45]. Although INH is a first-line drug that is used to treat tuberculosis, approximately 10% of INH-treated patients exhibit upregulated (≥3× the upper limit of normal [ULN]) activity of alanine transaminase (ALT), and approximately 1% of INH-treated patients exhibit even greater upregulated ALT activity (>5× ULN) [46]. INH is a prodrug that is metabolized to an acyl radical in *Mycobacterium tuberculosis* via bacterial catalase-peroxidase, and this acyl radical inhibits the synthesis of lipids in the cell wall [46]. Several enzymes are involved in the metabolism of INH in the mammalian liver, including *N*-acetyltransferase 2 (NAT2), amidases, and cytochrome P450 (CYP) enzymes [46,47,48,49]. Amidases metabolize INH and acetyl INH to hydrazine (Hz) and acetyl Hz, respectively. NAT2 acetylates INH, Hz, and acetyl Hz to generate acetyl INH, acetyl Hz, and diacetyl Hz, respectively. CYP enzymes metabolize acetyl Hz to generate reactive metabolites such as the acetylonium ion, acetyl radical, and ketene, which can bind covalently to macromolecules in hepatocytes to form adducts [46,47,48,49]. Previous studies have demonstrated that these metabolites and adducts (with the exception of diacetyl Hz) play roles in INH-induced hepatotoxicity via multiple mechanisms, including mitochondrial injury, oxidative stress, inflammation, and activation of abnormal immune responses [46,47,48,49]. These mechanisms are associated with hepatocyte apoptosis [1]. INH has been shown to induce the apoptosis of several human hepatic cell lines (HepG2, THLE-2, L02) [50,51,52] and in hepatocytes of a human organotypic culture model [53] and a rat liver [52]. In larval zebrafish that were treated with INH (6 mM for 72 h), significant increases were observed in both the expression of mRNAs that encode the pro-apoptotic markers Bax and cytochrome c and in ALT activity, suggesting that INH also induces hepatocyte apoptosis in zebrafish [30]. In this study, we demonstrated a significant increase in the ratio of fluorescence-emitting areas that were detected with narrow and wide filters using in vivo fluorescence imaging of Tg (pck1:Casper3GR) zebrafish treated with INH (6 mM for 24 h). These findings suggest that Tg (pck1:Casper3GR) is a suitable model for the assessment of the effect of target chemicals on hepatocyte apoptosis in a relatively easy and highly sensitive manner.

Several drugs have been identified as potential hepatoprotectants against INH-induced liver injury [54,55,56], such as UDCA [35]. UDCA suppresses the transcriptional activity of TP53, which is a key regulator of apoptosis, resulting in a decrease in the expression of the pro-apoptotic factor Bax and an increased expression of the anti-apoptotic factor Bcl-2 in rat hepatocytes [33,57]. We demonstrated that co-treatment of Tg (pck1:Casper3GR) with INH and UDCA significantly suppressed the increased ratio of fluorescence-emitting areas in zebrafish that were detected using narrow and wide filters. These findings suggest that UDCA suppresses INH-induced hepatocyte apoptosis.

The UDCA-mediated suppression of apoptosis also involves the activation of FXR, which is a nuclear receptor that positively regulates the expression of the primary bile-acid transporters ABCB11 (also known as Bsep) and ABCC2 (also known as Mrp2) on the canalicular membrane of hepatocytes [33]. As bile acid induces both the intrinsic and extrinsic pathways of apoptosis [1], ABCB11 and ABCC2 excrete toxic bile acid from hepatocytes. UDCA appears to suppress hepatocyte apoptosis by activating FXR and upregulating the ABCB11- and ABCC2-mediated excretion of bile acid. Consistent with this hypothesis, we demonstrated that the FXR agonist OCA significantly suppressed the increase in the ratio of fluorescence-emitting areas that were detected using narrow and wide filters in Tg (pck1:Casper3GR) that was treated with INH. Previous studies demonstrated that INH decreases the expression of FXR, ABCB11, and ABCC2 in both HepG2 cells and rat livers, and that pyridoxal isonicotinoyl hydrazone [58], which is generated by conjugation of INH with endogenous vitamin B6, is a potent FXR antagonist [37]. These findings suggest that INH causes hepatocellular cholestasis through inhibition of FXR activity, indicating that the stimulation of FXR could be a useful approach for ameliorating the effects of INH-induced liver injury.

Although a previous study demonstrated that DEX suppresses hepatocyte apoptosis in rats that were treated with both INH and LPS [59], we did not observe a significant protective effect of DEX with regard to hepatocyte apoptosis in Tg (pck1:Casper3GR) that was treated with INH alone. Corticosteroids are used empirically to treat DILI, but not all patients experience therapeutic effects [38,56]. In mice, DEX reportedly increases hepatocellular cholestasis via the suppression of FXR activity [60]. However, in the present study, DEX treatment did not increase the ratio of fluorescence-emitting areas that were detected using narrow and wide filters in Tg (pck1:Casper3GR) treated with INH, suggesting that DEX suppresses both inflammatory pathways and hepatocyte apoptosis. The present study involved only FRET-based imaging data of the Tg (pck1:Casper3GR) zebrafish. Further studies examining parameters such as the expression of pro- and anti-apoptosis markers, FXR, bile-acid transporters, and various inflammatory markers could help elucidate the mechanisms underlying INH-associated hepatotoxicity and the effects of UDCA, OCA, and DEX on INH-induced liver injury.

The ease of drug administration and FRET-based imaging makes our Tg (pck1:Casper3GR) zebrafish a powerful tool for use in studies to identify both hepatotoxic and hepatoprotective drugs. In silico analyses using public databases such as the adverse event reporting system and transcriptome data for tissues and/or cells exposed to toxicants could lead to efficient prediction of drugs and genes that may exert toxic or protective effects on a specific phenotype [55,61]. Various genome-editing technologies are also applicable to zebrafish. FRET-based imaging of Tg (pck1:Casper3GR) can be used to validate data from in silico predictions of drugs and genes that may be associated with hepatocyte apoptosis.

## 4. Materials and Methods

### 4.1. Ethics Statement

This study was approved by the Institutional Animal Care and Use Committee at Mie University (no. 2020-19). All animal experiments conformed to the ethical guidelines established by the committee.

### 4.2. Compounds

INH and DEX were purchased from Tokyo Chemical Industry (Tokyo, Japan). UDCA was purchased from Fujifilm Wako Pure Chemical (Osaka, Japan). OCA was purchased from MedChemExpress (Monmouth Junction, NJ, USA). Stock solutions of UDCA, OCA, and DEX were prepared in dimethyl sulfoxide (Fujifilm Wako Pure Chemical) to a concentration 1,000-fold higher than the final concentration used in the experiments. The stock solutions were then diluted 1,000-fold in 0.3× Danieau’s solution (19.3 mM NaCl, 0.23 mM KCl, 0.13 mM MgSO_4_, 0.2 mM Ca[NO_3_]_2_, 1.7 mM HEPES [pH 7.2]). The same final concentration of dimethyl sulfoxide (i.e., 0.1%) was used for the control in experiments using UDCA, OCA, or DEX. INH powder was dissolved in 0.3× Danieau’s solution to the final desired concentration. Dimethyl sulfoxide was not used in the control for experiments using only INH.

### 4.3. Zebrafish Husbandry

Zebrafish were maintained according to standard methods as described previously [62]. Briefly, zebrafish were raised at 28.5 ± 0.5°C with a 14 h/10 h light/dark cycle. We obtained embryos by natural mating and cultured the embryos in 0.3× Danieau’s solution until 7 dpf and then raised them in a recirculating system for housing zebrafish [63].

### 4.4. Generation of Tg (pck1:Casper3GR) Zebrafish

An albino zebrafish line [64] (Max Planck Institute for Developmental Biology, Tübingen, Germany) was used to generate the Tg zebrafish. The plasmid vector encoding Casper3GR was purchased from Evrogen (Moscow, Russia). The coding region of Casper3GR was cloned into the pT2AL200R150G vector [65] downstream of the ef1a promoter to generate the pT2-Casper3GR plasmid. The promoter region of zebrafish *pck1* (chr6:60,053,841-60,056,755, GRCz11/danRer11) was synthesized and cloned into pUC57 (GenScript, NJ, USA). The coding region of Casper3GR was amplified using specific primers (5′-gccaccatggtgagcgagctgattaagg-3′ and 5′-ctcgagccgggcccaagtgatctc-3′) and pT2-Casper3GR as the template. Inverse PCR was performed using specific primers (5′-tgggcccggctcgaggtttgcatgttctccctgtgttggcg-3′, 5′-gctcaccatggtggcgatgagcggagggatggagagatgct-3′) and pUC57-pck1 as the template. After digestion of the template plasmids using *Dpn*I (Takara Bio Inc., Shiga, Japan), the amplified PCR products were fused using an In-Fusion HD cloning kit (Takara Bio Inc.) to generate the pT2-pck1-Casper3GR plasmid. Transposase mRNA was synthesized using an mMessage mMachine SP6 transcription kit (Thermo Fisher Scientific, MA, USA) and pCS-zT2TP [66] as the template. To generate Tg (pck1:Casper3GR) zebrafish, pT2-pck1-Casper3GR plasmid and transposase mRNA were injected into zebrafish embryos at the 1- to 4-cell stage. The Tol2 sequences are recognized by the transposase, which integrates the insert between the Tol2 sequences in the zebrafish genome [67]. The larvae expressing Casper3GR in the liver were selected and maintained. Mature F0 single-Tg zebrafish were mated with albino zebrafish, and single-Tg F1 fish were selected based on the expression of Casper3GR in the liver and maintained. Mature F1 females and males were mated to generate F2 zebrafish. This step has been continually repeated to maintain the Tg (pck1:Casper3GR) zebrafish. In this study, we used Tg (pck1:Casper3GR) zebrafish larvae that strongly expressed Casper3GR in the liver, presumably due to the inheritance of transgenes from both parents. Tg (pck1:Casper3GR) will be available from National Bioresource Project Zebrafish (https://shigen.nig.ac.jp/zebra/index_en.html, accessed on 28 October 2021) or the authors.

### 4.5. Exposure of Tg (pck1:Casper3GR) to Chemicals and In Vivo Fluorescence Imaging

Mature Tg (pck1:Casper3GR) were mated, and the resulting embryos were transferred to 6-well plates (10 embryos per well) and treated with test chemicals at the indicated concentrations from 4 dpf to 4.5 or 5 dpf. After treatment, the larvae were transferred to fresh 0.3× Danieau’s solution containing 2-phenoxyethanol (500 ppm) for anesthesia and then transferred to a glass-bottom dish (Cell View Cell Culture Dish, 4 compartments, Greiner Bio-One, Austria) along with the medium. The zebrafish were then observed using an inverted fluorescence microscope (Axio Observer, Zeiss, Oberkochen, Germany) with the following filters: wide filter (Ex/Em 460–480/567–647 nm) and narrow filter (Ex/Em 460–480/505–530 nm). Fluorescent signals in the obtained images were quantified using the image processing package Fiji [68]. Briefly, 16-bit images were imported into Fiji, and the areas of wide and narrow fluorescence above the threshold (mean gray value > 4000) within the liver were measured. The ratios of narrow and wide areas above the thresholds were then calculated.

### 4.6. Caspase 3/7 Assay

Caspase 3/7 assay (Caspase-Glo 3/7 assay, Promega, WI, USA) was performed according to the manufacturer’s instructions. Briefly, zebrafish at 1 mpf were treated with or without INH (0.5 mM) for 24 h. After treatment, each zebrafish was anesthetized. The liver was removed from each zebrafish and transferred to a 96 well plate with white well walls (136101, Thermo Fisher Scientific). Caspase-Glo 3/7 Reagent (100 μL) was added to each well containing liver. The plates were placed on a shaker for 30 s and incubated at room temperature for 60 min. The luminescence of the mixtures was measured using a luminometer (LB941, Berthold, Württemberg, Germany).

### 4.7. Statistical Analysis

Shapiro–Wilk normality tests were used to examine the distribution of the data for each group. Because the test revealed that these data were not always normally distributed, we performed statistical analyses using the Mann–Whitney *U* test (for two groups) and Kruskal–Wallis test and Dunn’s multiple comparisons test (for more than two groups) in Prism 9 software (GraphPad, CA, USA) to examine the differences between groups.

## 5. Conclusions

Tg (pck1:Casper3GR), a transgenic zebrafish line developed in this study that dominantly expresses FRET probe Casper3GR in the liver at the larval stage, can be a powerful tool for identifying both hepatotoxic and hepatoprotective drugs as well as analyzing the effects of genes of interest on hepatocyte apoptosis using FRET-based in vivo imaging.

## Figures and Tables

**Figure 1 pharmaceuticals-14-01117-f001:**
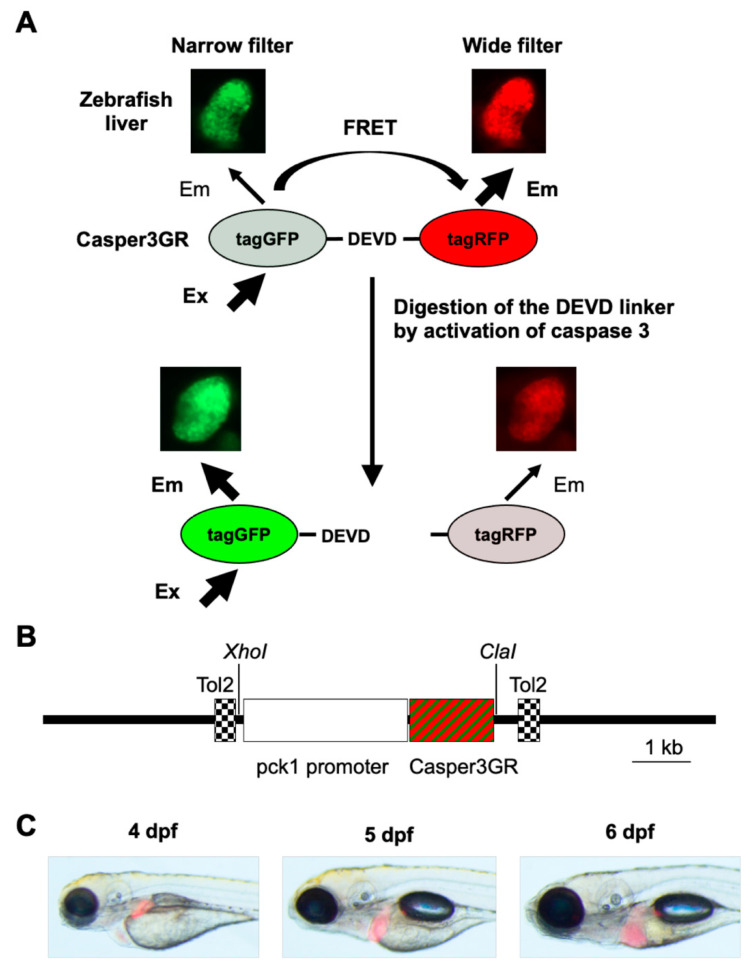
Assessment of hepatic apoptosis in zebrafish using FRET-based imaging. (**A**) FRET-based imaging technique developed in this study. Zebrafish expressing Casper3GR in hepatocytes were subjected to in vivo fluorescent imaging. When caspase 3 is not activated in the hepatocytes, the linker between tagGFP and tagRFP is not digested, resulting in FRET (i.e., excitation at the absorbance wavelength of tagGFP causes emission by tagRFP). When caspase 3 is activated in the hepatocytes, the linker is digested, which causes a decrease in FRET efficiency. In vivo fluorescence images of zebrafish liver obtained using a narrow filter (Ex/Em: 460–480/505–530 nm) and wide filter (Ex/Em: 460–480/567–647 nm) are also shown. (**B**) Diagram of the transposon vector used in this study. The coding region of Casper3GR was placed downstream of the regulatory sequences of pck1 and cloned between two Tol2 sequences in the vector backbone to direct selective protein expression in hepatocytes. (**C**) In vivo fluorescence imaging of Tg (pck1:Casper3GR) at 4, 5, and 6 dpf. The bright-field, GFP filter (Ex/Em: 460–490/510 nm), and RFP filter (Ex/Em: 545–580/610 nm) images are merged.

**Figure 2 pharmaceuticals-14-01117-f002:**
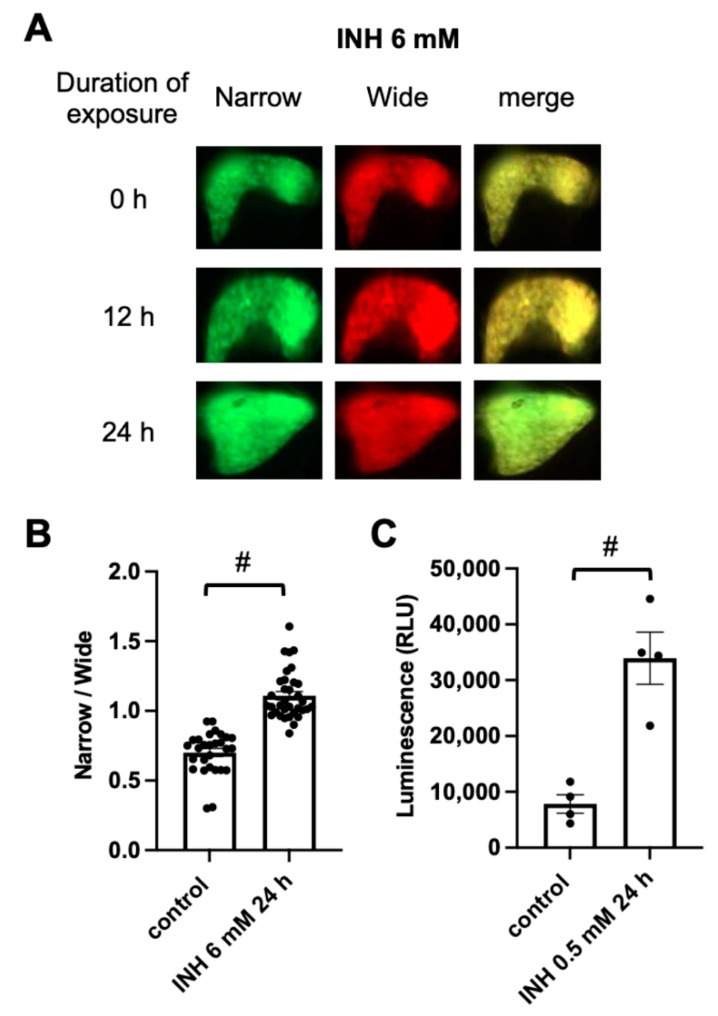
Assessment of hepatic apoptosis in INH-treated Tg (pck1:Casper3GR). (**A**) Representative in vivo fluorescence images of Tg (pck1:Casper3GR) treated with INH at 4 dpf. Images acquired using a narrow and wide filter and merged images are shown. (**B**) Hepatic apoptosis was assessed as the ratio of narrow and wide area of fluorescence above the threshold in the liver of zebrafish at 5 dpf treated with or without INH (6 mM for 24 h). Circles represent individual zebrafish, and bars represent the group means with standard error of the means. n = 28 and 34 for control and INH 6 mM 24 h, respectively. # *p* < 0.05 vs. control. (**C**) Hepatic apoptosis was assessed based on the luminescence derived from substrates that become luminogenic after digestion at the DEVD site by activated caspase 3/7 in extracts of liver isolated from zebrafish at 1 mpf treated with or without INH (0.5 mM for 24 h). Circles represent individual zebrafish, and bars represent the group means with standard error of the means. *n* = 4 for each group.

**Figure 3 pharmaceuticals-14-01117-f003:**
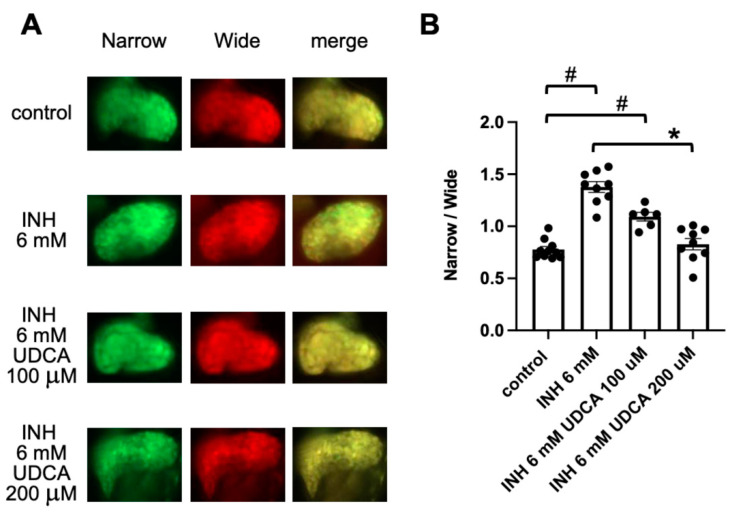
Assessment of hepatic apoptosis in Tg (pck1:Casper3GR) treated with INH with and without UDCA. (**A**) Representative in vivo fluorescence images of Tg (pck1:Casper3GR) treated with INH with and without UDCA from 4 to 5 dpf. Images acquired using narrow and wide filters and merged images are shown. (**B**) Hepatic apoptosis was assessed as the ratio of narrow and wide areas of fluorescence above the threshold in the liver of zebrafish at 5 dpf treated with INH with and without UDCA. Circles represent individual zebrafish, and bars represent the group means with standard error of the means. *n* = 10, 9, 6, and 9 for control, INH 6 mM, INH 6 mM UDCA 100 μM, and INH 6 mM UDCA 200 μM, respectively. # *p* < 0.05 vs. control, * *p* < 0.05 vs. INH 6 mM.

**Figure 4 pharmaceuticals-14-01117-f004:**
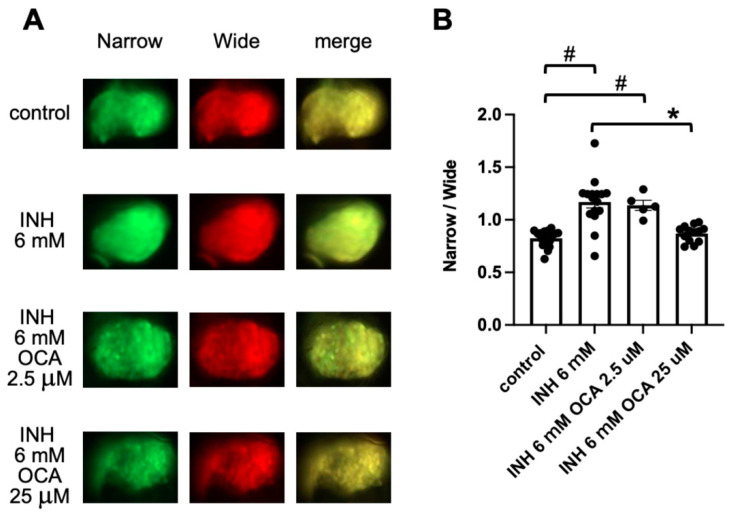
Assessment of hepatic apoptosis in Tg (pck1:Casper3GR) treated with INH with and without OCA. (**A**) Representative in vivo fluorescence images of Tg (pck1:Casper3GR) treated with INH with and without OCA from 4 to 5 dpf. Images acquired using narrow and wide filters and merged images are shown. (**B**) Hepatic apoptosis was assessed as the ratio of narrow and wide areas of fluorescence above the threshold in the liver of zebrafish at 5 dpf treated with INH with and without OCA. Circles represent individual zebrafish, and bars represent the group means with standard error of the means. n = 18, 16, 5, and 13 for control, INH 6 mM, INH 6 mM OCA 2.5 μM, and INH 6 mM OCA 25 μM, respectively. # *p* < 0.05 vs. control, * *p* < 0.05 vs. INH 6 mM.

**Figure 5 pharmaceuticals-14-01117-f005:**
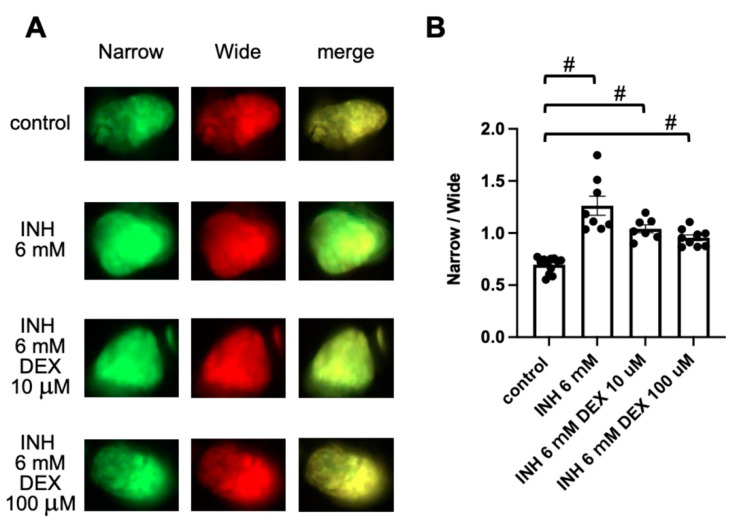
Assessment of hepatic apoptosis in Tg (pck1:Casper3GR) treated with INH with and without DEX. (**A**) Representative in vivo fluorescence images of Tg (pck1:Casper3GR) treated with INH with and without DEX from 4 to 5 dpf. Images acquired using narrow and wide filters and merged images are shown. (**B**) Hepatic apoptosis was assessed as the ratio of narrow and wide areas of fluorescence above the threshold in the liver of zebrafish at 5 dpf treated with INH with and without DEX. Circles represent individual zebrafish, and bars represent the group means with standard error of the means. n = 13, 8, 7, and 9 for control, INH 6 mM, INH 6 mM DEX 10 μM, and INH 6 mM DEX 100 μM, respectively. # *p* < 0.05 vs. control.

## Data Availability

Data is contained within the article.

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
