# Peer review of "Generation of a Transgenic Zebrafish Line for In Vivo Assessment of Hepatic Apoptosis"

_pharmaceuticals, 2021, doi:10.3390/ph14111117_

Round 1

Reviewer 1 Report

The manuscript entitled “Generation of a transgenic zebrafish line for in vivo assessment of hepatic apotosis” by Aina Higuchi and coworkers is the revised resubmission. The revised manuscript is clear, well presented and the authors responded to all my previous concerns. As it stands, the manuscript has been significantly improved. 

Reviewer 2 Report

None

This manuscript is a resubmission of an earlier submission. The following is a list of the peer review reports and author responses from that submission.

Round 1

Reviewer 1 Report

Higuchi et al have developed an exciting line of transgenic zebra fish that express an elegant dual-fluorescent FRET marker in hepatocytes that is cleaved by endogenous Casp3 under apoptosis-inducing conditions. In vivo cleavage de-tethers the green- and red-fluorescent proteins, and quantitatively decreases FRET under GFP-excitation illumination.  They then present clear data of apoptosis readouts in the presence of apoptosis-inducing conditions (INH treatment) or co-administration of an apoptosis-inducing and a cytoprotective (DEX) agent.  Overall, this is a clearly written high quality manuscript that I hope will be published with high priority.  However, prior to publication, I feel two deficiencies need to be addressed.

1) Despite the elegance and demonstrated utility of the transgenic zebra fish line reported, this line, in fact, only has scientific value if it is made available to the greater scientific community. I found no statement indicating that the fish line has been submitted to a public repository or other statement of availability to the scientific community.  The paper should only be published if it includes an adequate statement ensuring availability to the research community.  Indeed for this reason, in the checklist above, I gave "no answer" for Overall Merit.  However, if this line is made available to the research community, the study will have high overall merit.

2) The methods are excessively terse and incomplete. A figure diagramming the transposon vector that was used; validation of genotype, copy-number, inheritance of genotype and expression levels; and integration site(s) mapping should be included.  This figure, along with improved descriptions in the Methods section (e.g., which transposon/transposase system was used?) should allow the reader to understand what was done, evaluate this fully, and replicate the study, as desired.

Reviewer 2 Report

The manuscript entitled “Generation of a transgenic zebrafish line for in vivo assessment of hepatic apotosis” by Aina Higuchi and coworkers describes a transgenic zebrafish line expressing a pck1:Casper3GR construct allowing to monitor caspase 3 activity in liver based on FRET measurements. This transgenic line is validated by showing that isoniazid induces hepatocyte apoptosis while ursodeoxycholic acid and obeticholic acid exert hepatoprotection. Furthermore, the authors used this zebrafish line to demonstrate an hepatotoxicity effect of dexamethasone.

The manuscript is clear, well presented and comprehensively written. The experiments are well conducted and the conclusions supported by the data presented. However, the authors should consider the following points:

(i) Since phosphoenolpyruvate carboxykinase gene expression is hepato-enriched but not strictly hepato-specific the authors could emphasize on the choice of using the pck1 promoter in their construct instead of fabp10a for example. Is the pck1:Casper3GR expression detected in other tissues such as kidney or muscle?

(ii) Figure 1, as far as I understand, 0 h corresponds to zebrafish imaging at 4 dpf, 12 h to zebrafish imaging at 4.5 dpf and 24 h to zebrafish imaging at 5 dpf. A better control should be to treat zebrafish larvae with DMSO at 4 dpf and image at 5 dpf. Could the authors clarify this point?

(iii) The authors seem to use no treatment as controls in their experiments. Since all the compounds (INH, DEX, UDCA and OCA) are dissolved in DMSO, a real control should be the treatment of larvae with DMSO. Does DMSO modulate caspase 3 activity in vivo?

(iv) The authors should indicate the number of measurements (n = ?) performed in their experiments (Fig. 2, 3, 4 and 5).

(v) The authors used the symbols * and # in Fig. 3B, 4B and 5B, but the legends are not always clear. Could the authors use brackets to indicate which condition is significantly different to which one in the figures? In addition, Legend of Fig. 5B specify “*p<0.05 vs INH 6 mM and DEX 0 µM” but there is no asterisk in the figure. Could the authors clarify this point?

Reviewer 3 Report

Higuchi et al. generate a new zebrafish transgenic line for in vivo assessment of hepatic apoptosis. The zebrafish line optimized in the paper could be a tool for live imaging finalized to the study of hepatic apoptosis in zebrafish. Nevertheless, although the authors presented a good tool for toxicological studies, they do not take into account the toxicity of the compounds used to validate the model. In fact, authors used molecules hepatoprotective (UDCA and OCA) or hepatotoxic (DEX) to validate their model without  literature data or original experiments finalized to support embryos tollerance to the treatments. I suggest to check and validate toxicity of the molecules used in the study.